

# Biomechanical, physiological and anthropometrical predictors of performance in recreational runners

Leonardo A. Peyré-Tartaruga[1,2,*], Esthevan Machado[2,*],
Patrick Guimarães[2], Edilson Borba[2,3], Marcus P. Tartaruga[3,4],
Cosme F. Buzzachera[1], Luca Correale[1], Fábio Juner Lanferdini[2,5] and
Edson Soares da Silva[2,6]

[1] Human Performance Laboratory (LocoLab), Department of Public Health, Experimental Medicine and Forensic Sciences, University of Pavia, Pavia, Italy
[2] LaBiodin, Biodynamics Laboratory, Universidade Federal do Rio Grande do Sul, Porto Alegre, Rio Grande do Sul, Brazil
[3] Postgraduate Program in Physical Education, Universidade Federal do Paraná, Curitiba, Brazil
[4] Department of Physical Education, Universidade Estadual do Centro Oeste do Paraná, Guarapuava, Brazil
[5] Biomechanics Laboratory, Universidade Federal de Santa Maria, Santa Maria, Brazil
[6] Inter-university Laboratory of Human Movement Biology, UJM-Saint-Etienne, Saint-Etienne, France
* These authors contributed equally to this work.

Corresponding author
Leonardo A. Peyré-Tartaruga,
leonardo.tartaruga@ufrgs.br

## ABSTRACT

**Background:** The maximal running speed ($V_{MAX}$) determined on a graded treadmill test is well-recognized as a running performance predictor. However, few studies have assessed the variables that predict $V_{MAX}$ in recreationally active runners.

**Methods:** We used a mathematical procedure combining Fick's law and metabolic cost analysis to verify the relation between (1) $V_{MAX}$ versus anthropometric and physiological determinants of running performance and, (2) theoretical metabolic cost versus running biomechanical parameters. Linear multiple regression and bivariate correlation were applied. We aimed to verify the biomechanical, physiological, and anthropometrical determinants of $V_{MAX}$ in recreationally active runners. Fifteen recreationally active runners participated in this observational study. A Conconi and a stead-steady running test were applied using a heart rate monitor and a simple video camera to register the physiological and mechanical variables, respectively.

**Results:** Statistical analysis revealed that the speed at the second ventilatory threshold, theoretical metabolic cost, and fat-mass percentage confidently estimated the individual running performance as follows: $V_{MAX}$ = 58.632 + (−0.183 * fat percentage) + (−0.507 * heart rate percentage at second ventilatory threshold) + (7.959 * theoretical metabolic cost) ($R^2$ = 0.62, $p$ = 0.011, RMSE = 1.50 km.h$^{-1}$). Likewise, the theoretical metabolic cost was significantly explained ($R^2$ = 0.91, $p$ = 0.004, RMSE = 0.013 a.u.) by the running spatiotemporal and elastic-related parameters (contact and aerial times, stride length and frequency, and vertical oscillation) as follows: theoretical metabolic cost = 10.421 + (4.282 * contact time) + (−3.795 * aerial time) + (−2.422 * stride length) + (−1.711 * stride frequency) + (0.107 * vertical oscillation).

**Conclusion:** Critical determinants of elastic mechanism, such as maximal vertical force and vertical and leg stiffness were unrelated to the metabolic economy. $V_{MAX}$, a valuable marker of running performance, and its physiological and biomechanical determinants can be effectively evaluated using a heart rate monitor, treadmill, and a digital camera, which can be used in the design of training programs to recreationally active runners.

## INTRODUCTION

Recreational runners engage in the activity with various objectives, ranging from participation in long-distance running events to the improvement of physical conditioning (*Scheerder, Breedveld & Borgers, 2015*). These objectives can be achieved through changes in biomechanical, physiological, and anthropometrical factors (*Foster & Lucia, 2007*; *Novacheck, 1998*; *Saunders et al., 2004*), which may lead to improved running performance (*Saunders et al., 2004*; *Foster & Lucia, 2007*; *Tartaruga et al., 2012*). However, beyond these primary factors, secondary factors such as anatomical factors (*e.g.*, lower limb size) can also play crucial role in determining running economy, which is clearly important for overall running performance (*Foster & Lucia, 2007*).

Conceptual models present these variables directly or indirectly (*Bassett & Howley, 2000*; *Saunders et al., 2004*). Biomechanical factors, for example, are associated to movement patterns that possibly play a significant role in running performance, such as maximal vertical force, vertical stiffness ($K_{VERT}$), leg stiffness ($K_{LEG}$), stride frequency, stride length, contact time, aerial time, and vertical oscillation of the center of mass (*Boullosa et al., 2020*; *Gómez-Molina et al., 2017*; *Saunders et al., 2004*). Physiological and anthropometrical factors, such as heart rate corresponding to the onset of blood lactate accumulation, maximal running speed ($V_{MAX}$), speed associated with the second ventilatory threshold ($V_{2VT}$), body mass, and fat percentage, can also be determinants of performance (*Abe et al., 1999*; *Boullosa et al., 2020*; *Daniels, 2013*; *Gómez-Molina et al., 2017*).

The variable with a strong predictive power for running performance is $V_{MAX}$ (*Gómez-Molina et al., 2017*; *Houmard et al., 1991*; *Noakes, Myburgh & Schall, 1990*; *Scott & Houmard, 1994*; *Stratton et al., 2009*). *Stratton et al. (2009)* demonstrated that the $V_{MAX}$ is strongly correlated with long-distance running performance in both trained and untrained individuals. Similarly, *Scott & Houmard (1994)* observed this association between $V_{MAX}$ and running performance among trained men and women. Furthermore, the $V_{MAX}$ has been utilized as a critical performance marker for trained runners (*Lanferdini et al., 2020*). Current literature has shown that somatic and exertion variables, including age, body mass index, maximum oxygen uptake, blood lactate concentration at anaerobic threshold, and pulmonary ventilation, accurately estimate $V_{MAX}$ in more than 4,000 endurance runners (*Wiecha et al., 2022*). However, many studies have been limited to either trained or elite

athletes, thus limiting the generalizability of findings regarding the specific variables determining $V_{MAX}$ in recreationally active runners through low-cost tests with simple instrumentation (*Abe et al., 1999*; *Gómez-Molina et al., 2017*; *Lanferdini et al., 2020*; *McLaughlin et al., 2010*). Further, very simple measures as body mass, fat mass, previous performance data of training and races may give important clues on performance prediction in distance runners (*Melo et al., 2018*, *2022*). Allometric models have also been applied to understand the relationship between biomechanics versus running economy and performance (*Detoni et al., 2015*; *Tartaruga et al., 2013*).

Measurements of biomechanical, physiological, and anthropometrical variables can be easily determined through low-cost field tests, such as the maximal incremental test and the rectangular running test on a treadmill, based on known protocols for monitoring the individuals' heart rate and recording gait patterns (*Marfell-Jones, Stewart & De Ridder, 2012*; *Sentija, Vucetic & Markovic, 2007*; *Morin et al., 2005*). These tests enable the estimation of athletes' maximal aerobic capacity and the observation of their mechanical running patterns during the test, as shown by many studies that performed those respective tests (*Tartaruga et al., 2012*, *2013*; *Lanferdini et al., 2020*). As a result, professional coaches and researchers can use this information to estimate running performance by determining $V_{MAX}$ (*Stratton et al., 2009*). In addition, there are approaches that utilize maximal incremental tests to evaluate running economy (*di Prampero et al., 2009*). Surprisingly, little information exists on relationship between elastic mechanism variables and metabolic cost in recreational runners (*Peyré-Tartaruga et al., 2021*). Further, previous study has shown that the elastic mechanism plays a crucial role in the relationship between metabolic cost and speed of running in trained runners (*Carrard, Fontana & Malatesta, 2018*).

Therefore, we aimed to investigate the biomechanical, physiological, and anthropometrical determinants of $V_{MAX}$ in recreationally active runners. We hypothesized that anthropometrical factors, such as body mass and fat percentage, along with the physiological variable $V_{2VT}$ would emerge as the primary predictors of $V_{MAX}$, given the high heterogeneity observed among recreationally active runners. Further, our second objective was to relate biomechanical variables of running versus metabolic cost. Thus, we also hypothesized that mechanical variables related to the elastic mechanism would be associated with the metabolic cost. This hypothesis is based on the findings of da Rosa and coworkers, who clearly demonstrated landing-takeoff asymmetries of running optimized in runners with good performance in comparison to runners with lower performance (*da Rosa et al., 2019*).

## MATERIALS AND METHODS

### Participants

The study was approved by the Research Ethics Committee of Universidade Federal do Rio Grande do Sul (Protocol number 2.437.616), according to the Declaration of Helsinki. Fifteen recreationally active runners (age 30.7 ± 7.3; height 167.2 ± 8; body mass 67.4 ± 11.3) participated in the present study (see Table 1), six males and nine females. Participants were classified as recreationally active based on their engagement in at least

**Table 1 Values of mean, standard deviation (SD) and confidence interval (CI) of anthropometrical, physiological and biomechanical variables in recreationally active runners.**

| Variable | Female ($n$ = 9) Mean/SD | Male ($n$ = 6) Mean/SD | Total ($n$ = 15) Mean/SD | CI (95%) |
|---|---|---|---|---|
| Age (years) | 30.2 ± 6.9 | 31.3 ± 8.5 | 30.7 ± 7.3 | [27–34.3] |
| Height (cm) | 162.6 ± 6.5 | 174.2 ± 4.2 | 167.2 ± 8 | [163.2–171.3] |
| L (m) | 0.88 ± 0.05 | 0.92 ± 0.02 | 0.90 ± 0.04 | [0.87–0.92] |
| ΔL (m) | 0.09 ± 0.02 | 0.10 ± 0.02 | 0.09 ± 0.02 | [0.08–0.10] |
| Body mass (kg) | 63.7 ± 9.0 | 73.0 ± 12.9 | 67.4 ± 11.3 | [61.7–73.1] |
| Fat percentage (%) | 25.2 ± 4.8 | 16.4 ± 5.2 | 21.7 ± 6.5 | [18.4–25] |
| $HR_{2VT}$ (bpm) | 185.1 ± 6.9 | 174.5 ± 11.5 | 180.9 ± 10.2 | [175.7–186] |
| $HR_{MAX}$ (bpm) | 192.1 ± 6.7 | 185.5 ± 11.8 | 189.5 ± 9.3 | [184.7–194.2] |
| $V_{2VT}$ (km.h$^{-1}$) | 12.6 ± 1.7 | 13.9 ± 1.3 | 13.1 ± 1.7 | [12.26–13.96] |
| $V_{MAX}$ (km.h$^{-1}$) | 13.9 ± 2.0 | 15.8 ± 2.1 | 14.7 ± 2.2 | [13.56–15.75] |
| Cost_theor | 1.07 ± 0.05 | 1.06 ± 0.04 | 1.07 ± 0.04 | [1.044–1.086] |
| CT (s) | 0.26 ± 0.02 | 0.28 ± 0.03 | 0.27 ± 0.03 | [0.26–0.28] |
| AT (s) | 0.09 ± 0.03 | 0.09 ± 0.02 | 0.09 ± 0.03 | [0.08–0.11] |
| SL (m) | 1.93 ± 0.13 | 1.97 ± 0.16 | 1.95 ± 0.14 | [1.87–2.02] |
| SF (Hz) | 2.89 ± 0.19 | 2.84 ± 0.24 | 2.87 ± 0.21 | [2.77–2.97] |
| Δy (cm) | 9.8 ± 1.2 | 10.6 ± 1.5 | 10.8 ± 2.9 | [9.46–10.84] |
| $F_{MÁX}$ (N.kg$^{-1}$) | 1.4 ± 0.2 | 1.5 ± 0.2 | 1.4 ± 0.2 | [1.31–1.53] |
| $K_{LEG}$ (N.m$^{-1}$) | 16.1 ± 4.2 | 16.1 ± 3.1 | 16.1 ± 3.7 | [14.23–17.96] |
| $K_{VERT}$ (N.m$^{-1}$) | 16.3 ± 4.3 | 16.2 ± 3.1 | 16.2 ± 3.7 | [14.36–18.13] |

**Note:**
L, leg length; ΔL, leg deformation; $HR_{2VT}$, heart rate associated to second ventilatory threshold; $HR_{MAX}$, maximal heart rate; $V_{2VT}$, speed associated with the second ventilatory threshold; $V_{MAX}$, maximal running speed; Cost_theor, theoretical metabolic cost; CT, contact time; AT, aerial time; SL, stride length; SF, stride frequency; Δy, vertical oscillation of the center of mass; $F_{MAX}$, maximal vertical force; $K_{LEG}$, leg stiffness; $K_{VERT}$, vertical stiffness.

150 to 300 min of moderate-intensity running training or 75 to 150 of vigorous-intensity running training per week, along with muscle-strengthening exercises for two or more days a week as complementary training (*McKay et al., 2022*). The inclusion criteria for participation in this study included being free of chronic joint pain, musculoskeletal or bone injuries over the 6-month period preceding the experiments. All participants were part of the running extension project of the Research Group LOCOMOTION—Mechanics and Energetics of Terrestrial Locomotion. They were informed about the risks and benefits of the study, and voluntarily signed an informed consent form agreeing to participate in this study.

## Experimental design

This is a cross-sectional study following the recommendations of the STROBE checklist (*von Elm et al., 2014*). All data were collected in a single day for each subject. At the first moment of the visit, body mass, height, and the data needed to calculate the fat percentage were collected. Subsequently, all subjects performed a 5-min warm-up at 10 km.h$^{-1}$ before conducting the maximal incremental test protocol on a treadmill (Super ATL Inbrasport-

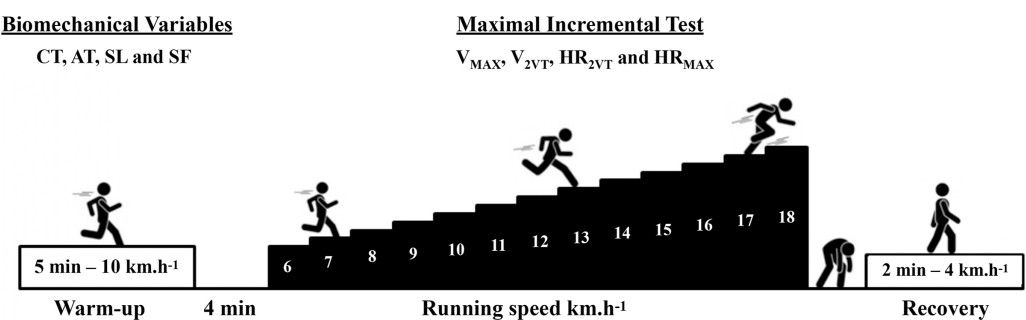

**Figure 1** **Experimental design.** Maximal incremental test (*Sentija, Vucetic & Markovic, 2007*). Maximal running speed (V$_{MAX}$), speed associated with the second ventilatory threshold (V$_{2VT}$), heart rate associated with the second ventilatory threshold (HR$_{2VT}$), maximal heart rate (HR$_{MAX}$), contact time (CT), aerial time (AT), stride length (SL) e stride frequency (SF).

Inbramed, Porto Alegre, Brazil). Thirty seconds of running in the sagittal plane were recorded in the 4$^{th}$ min of warm-up on the treadmill using a digital camera (Casio EX-ZR1000, 120 Hz; Casio, Tokyo, Japan) for biomechanical data analysis. The main part of the maximal incremental test started at 6 km.h$^{-1}$, with an increase of 1 km.h$^{-1}$ per minute until the participants reached exhaustion. Heart rate was recorded every 10 s during the maximal incremental test (Polar FT1, Kempele, Finland) to estimate heart rate associated to second ventilatory threshold (HR$_{2VT}$) and maximal heart rate (HR$_{MAX}$). Rating of perceived effort (RPE) (1 to 10 items) was collected at the end of each test stage (*Borg, 1982*).

To minimize the impact of confounding factors, the participants utilized their regular running shoes throughout the analysis. Furthermore, all subjects involved in the study typically underwent treadmill evaluations as part of their routine every 3 months.

## Anthropometry

Body mass (kg) was measured using a portable electronic scale (model UP-150; Urano, São Paulo, Brazil) with a resolution of 100 g. Height, body perimeters and the sum of skinfolds were assessed using a tape measure and a caliper, both with resolutions of 1 mm (CESCORF, Porto Alegre, Brazil), to determine the fat percentage according to the International Society for Advancement of Kinanthropometry (*Marfell-Jones, Stewart & De Ridder, 2012*). All data collections were performed by an experienced technician, independent of the research.

## Maximal incremental test

The protocol began with a 5-min warm-up at 10 km.h$^{-1}$ on a treadmill set at a fixed 1% inclination. Participants had a 4-min rest between the warm-up and the main test. The initial speed of the test was 6 km.h$^{-1}$, with an additional 1 km.h$^{-1}$ added each minute until participants reached exhaustion. The RPE was assessed at the end of each stage (*Sentija, Vucetic & Markovic, 2007*). The heart rate was recorded every 10 s during the incremental test. The V$_{MAX}$ and HR$_{MAX}$ were determined by the highest speed and heart rate achieved during a complete running stage (60 s), respectively. If the last stage was not

completed, the $V_{MAX}$ and $HR_{MAX}$ would be determined by the speed and HR of the previous complete stage. Subsequently, the runners were instructed to walk for 2 min at 4 km.h$^{-1}$ for recovery (see Fig. 1).

Using the HR data recorded every 10 s by a sensor (Polar FT1, Kempele, Finland) during the maximal incremental test, the heart rate deflection point for each individual was identified in order to estimate the $V_{2VT}$ and the $HR_{2VT}$ (*Sentija, Vucetic & Markovic, 2007*). The $HR_{MAX}$, $HR_{2VT}$, $V_{MAX}$ and $V_{2VT}$ data obtained during the maximal incremental test were then used to calculate the theoretical metabolic cost.

## Biomechanical variables

The biomechanical variables were assessed using a digital camera (Casio EX-ZR1000, 120 Hz; Casio, Tokyo, Japan). The foot landing and take-off events on the ground were identified during ten steps in the Kinovea software version 0.8.15 at a constant speed of 10 km.h$^{-1}$. Regarding to the validity of the Kinovea software, differences of 0.83, 2.02 and −1.19 degrees were observed for hip, knee, and ankle angles, respectively, when compared to 3D motion analysis software. In addition, it was observed a good correlation in terms of intra-rater reliability over two sessions (ICC > 0.85) for hip, knee, and ankle angles (*Fernández-González et al., 2020*).

After the event identification, the contact time (s), aerial time (s), stride length (m) and stride frequency (Hz) were determined (*Tartaruga et al., 2012*). The maximal vertical force ($F_{MAX}$) was calculated using Eq. (1). From this variable, the vertical oscillation of the center of mass (Δy) was estimated using the equations proposed by *Morin et al. (2005)*:

$$F_{MAX} = m \cdot g \cdot \frac{\pi}{2} \cdot \left( \frac{AT}{CT} + 1 \right) \tag{1}$$

$$\Delta y = \frac{F_{MAX}}{m} \cdot \frac{CT^2}{\pi^2} + g \cdot \frac{CT^2}{8} \tag{2}$$

where $m$ is the body mass of the runners (kg), $g$ the gravitational acceleration (m.s$^2$), AT and CT are aerial time (s) and contact time (s), respectively.

Moreover, the vertical stiffness ($K_{VERT}$) and the leg stiffness ($K_{LEG}$) were also calculated using the equations proposed by *Morin et al. (2005)*:

$$K_{VERT} = \frac{F_{MAX}}{\Delta y} \tag{3}$$

$$K_{LEG} = \frac{F_{MAX}}{\Delta L} \tag{4}$$

where $\Delta L$ is the leg deformation (m), represented by the following equation:

$$\Delta L = L \cdot \sqrt{L^2 \cdot \left( \frac{v \cdot CT}{2} \right)^2 + \Delta y} \tag{5}$$

where $v$ is the treadmill speed (m.s$^{-1}$) and $L$ is the leg length of the runners (m) measured in standing (*Morin et al., 2005*).

## Theoretical metabolic cost

In sport settings, heart rate has been used for evaluating the effort conditions in several modalities, such as running (*Swain et al., 1998*; *Castagna, Krustrup & Póvoas, 2022*), cycling (*Guimarães et al., 2019*), swimming (*Hauber, Sharp & Franke, 1997*), and Nordic walking (*Monteiro et al., 2017*). However, a drawback of heart rate-based methods is the variability influenced by age, sex, BMI, and fitness level (*Hiilloskorpi et al., 1999*). For a better control of these confounding factors, the literature has used the percentage of heart rate reserve (% HRR), which is based on the resting and maximum heart rate, demonstrating a strong association with oxygen uptake (*Cunha, Farinatti & Midgley, 2011*). So, the % HRR has been proposed as a predictor of energy expenditure. Also, % HRR was accordingly validated as a proxy for oxygen uptake in laboratory settings for endurance exercises (*Guimarães et al., 2019*).

Thus, in our study, the % HRR was associated with $HR_{2VT}$ and considered a proxy of oxygen consumption. Additionally, this value was divided by the $V_{2VT}$, converting the representative value of metabolic power to metabolic cost with units relative to energy expended per unit distance traveled (*Peyré-Tartaruga & Coertjens, 2018*), here referred as theoretical metabolic cost (arbitrary units, a.u.):

$$COST_{theoretical} = \left( \frac{HR_{2VT}}{HR_{MAX}} \cdot 100 \right) \bigg/ \left( \frac{V_{2VT}}{V_{MAX}} \cdot 100 \right) \tag{6}$$

The dataset is available in https://doi.org/10.6084/m9.figshare.23912724.

## Statistical analysis

The data were presented as means, standard deviations and confidence intervals. The Shapiro-Wilk test was used to assess the data normality. Initially, backward multiple linear regression was performed to determine the relationship between the independent variables (body mass (kg), fat percentage (%), $HR_{2VT}$ (bpm), $HR_{MAX}$ (bpm), theoretical metabolic cost (a.u.), and $V_{2VT}$ (km.h$^{-1}$)) and the dependent variable, $V_{MAX}$ (km.h$^{-1}$). Subsequently, another backward multiple linear regression was applied to verify the relationship between the independent variables (contact time (s), aerial time (s), stride length (m), stride frequency (Hz), vertical oscillation of the center of the mass (cm), maximal vertical force (N.kg$^{-1}$), $K_{LEG}$ (N.m$^{-1}$), and $K_{VERT}$ (N.m$^{-1}$)) and the dependent variable, theoretical metabolic cost (a.u.). Expecting moderate or high collinearity between certain pairs of regressors due to deterministic relation between some variables, collinearity tests were performed and redundant factors were removed (body mass and $V_{2VT}$ were removed in the $V_{MAX}$ model; and maximal vertical force was removed in the theoretical metabolic cost model). In the collinearity statistics, tolerance and variance inflation factor test the assumption of multicollinearity. As a rule of thumb if variance inflation factor >10 and tolerance <0.1 the assumptions have been greatly violated. If the average variance inflation factor >1 and tolerance <0.2 the model may be biased. Variance inflation factor <1 and tolerance >0.2 confirms the model assumptions. Also, violation of assumptions was checked using Durbin-Watson test. All analyzes were performed using

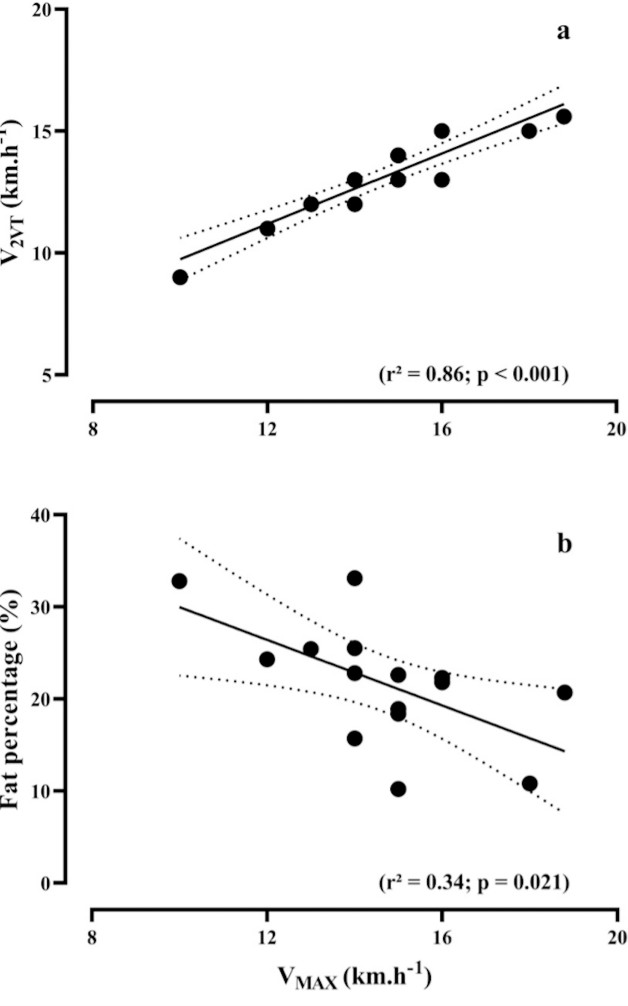

**Figure 2 Bivariate correlations.** Relationship between maximal running speed ($V_{MAX}$) during the maximal incremental test, *vs* speed associated with the second ventilatory threshold ($V_{2VT}$), (A) and fat percentage (B).                                                            

the statistical software Jasp (Version 0.16.2; Jasp, Amsterdam, Netherlands) for Mac. The significance level adopted was $\alpha = 0.05$.

## RESULTS

Table 1 shows the biomechanical, physiological and anthropometrical recreationally active runners variables in mean, standard deviation and 95% confidence interval.

The backward regression model for $V_{MAX}$ resulted in a highly significant model (F $(2,12) = 8.995$, $R^2 = 0.62$, $p = 0.011$) and a regression equation of

$$V_{MAX} = 58.632 + (-0.183 * \text{fat percentage}) + (-0.507 * \text{HR2VT\_perc}) \\ + (7.959 * \text{Cost\_theor}) \tag{7}$$

where HR2VT_perc is the heart rate as a percentage of $HR_{MAX}$, and Cost_theor is the theoretical metabolic cost. Specifically, three significant regression coefficients were found: fat percentage ($b_1 = -0.183$, $t = -2.981$, $p = 0.013$), heart rate as a percentage of $HR_{MAX}$

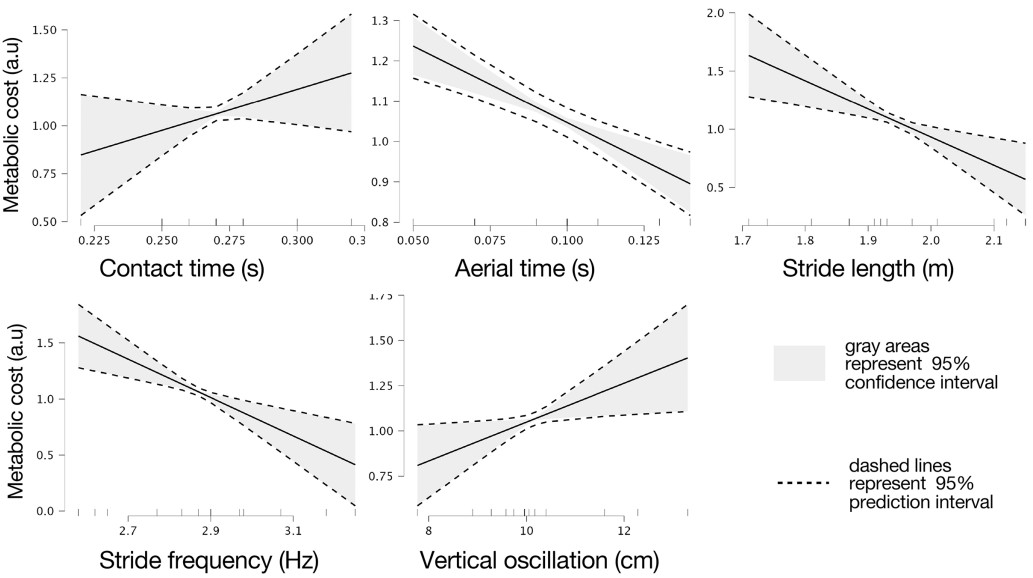

**Figure 3 Marginal effects of biomechanic variables on theoretical metabolic cost of running.** Marginal effects of contact and flight times, stride length and frequency, and vertical oscillation on theoretical metabolic cost of running based on intervals of prediction (dashed lines) and intervals of confidence (gray areas) at 95%.

($b_2 = -0.507$, $t = 2.464$, $p = 0.031$), and theoretical metabolic cost ($b_3 = 7.959$, $t = 0.841$, $p = 0.042$).

Similarly, the backward regression model for theoretical metabolic cost also resulted in a highly significant model ($F(5,9) = 15.936$, $R^2 = 0.91$, $p = 0.004$) and a regression equation as follows

$$
\begin{aligned}
\text{Cost\_theor} = 10.421 &+ (4.282 * contact\ time) + (-3.795 * aerial\ time) \\
&+ (-2.422 * stride\ length) + (-1.711 * stride\ frequency) \qquad (8) \\
&+ (0.107 * vertical\ oscillation)
\end{aligned}
$$

where Cost_theor is the theoretical metabolic cost. Specifically, five significant regression coefficients where found, contact time ($b_1 = 4.282$, $t = 1.780$, $p = 0.006$), aerial time ($b_2 = -3.795$, $t = 6.221$, $p = 0.002$), stride length ($b_3 = -2.422$, $t = 4.142$, $p = 0.009$), stride frequency ($b_4 = 1.711$, $t = 4.551$, $p = 0.006$), and vertical oscillation ($b_5 = 0.107$, $t = 2.963$, $p = 0.031$).

Figures 2A and 2B show the relationship between $V_{MAX}$ versus $V_{2VT}$ ($R^2 = 0.86$, $p < 0.001$) and fat percentage ($R^2 = 0.34$, $p = 0.021$), respectively.

Marginal effects of biomechanical variables on theoretical metabolic cost are presented in Fig. 3.

## DISCUSSION

The purposes of this study were twofold. First, we sought to verify which physiological ($HR_{2VT}$, $HR_{MAX}$, theoretical metabolic cost, and $V_{2VT}$) and anthropometrical variables (body mass, fat percentage) are determinant of $V_{MAX}$ in recreationally active runners.

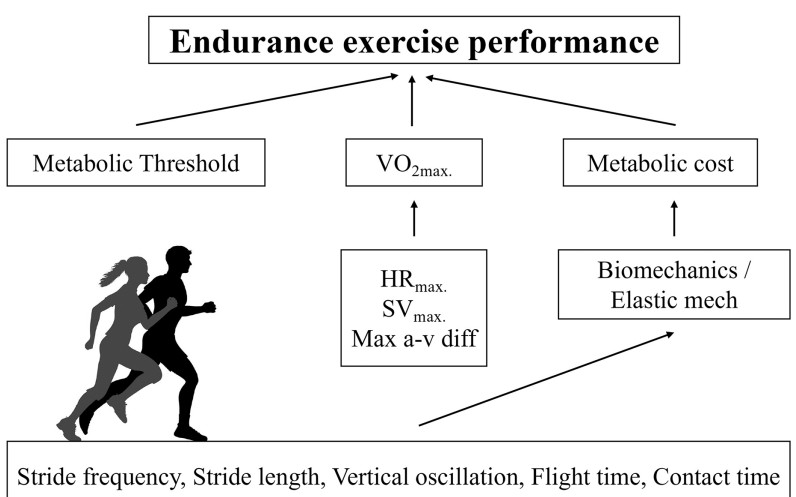

**Figure 4 Conceptual model for predicted $V_{MAX}$ in recreational runners.** Physiological determinants of running performance and the respective biomechanical determinants of metabolic cost of running.

Also, we sought to relate biomechanical ($K_{VERT}$, $K_{LEG}$, maximal vertical force, contact time, aerial time, stride length, stride frequency and vertical oscillation of the center of mass) variables are determinant of metabolic cost in recreationally active runners. Therefore, this is the first study to our knowledge to demonstrate specific physiological ($V_{2VT}$) and anthropometrical (fat percentage) variables determining $V_{MAX}$ in recreationally active runners (Fig. 4). We found that velocity associated with second ventilatory threshold, metabolic cost and fat percentage were predictors of $V_{MAX}$. Furthermore, our study provided new insights about the mechanical determinants of metabolic cost of running. We found that simple spatiotemporal variables and vertical oscillation were identified as determinants of the metabolic cost. Interestingly, the elastic mechanisms were not related to metabolic cost. These results reinforce the understanding that estimates of the elastic mechanism based on temporal parameters of running do not appear to be sensitive, as already observed in master athletes (*Pantoja et al., 2016*; *Cavagna, Legramandi & Peyré-Tartaruga, 2008*). On the other hand, the role of vertical oscillation in the metabolic cost of running is in line with previous findings in highly trained endurance runners (*Tartaruga et al., 2012*).

According to our results, $V_{2VT}$, theoretical metabolic cost, and fat-mass percentage collectively accounted for 62% of the total variance in $V_{MAX}$. These findings agree with several studies performed with recreational, trained and elite runners (*Noakes, Myburgh & Schall, 1990*; *Zacharogiannis & Farrally, 1993*; *Abe et al., 1999*; *Bassett & Howley, 2000*; *Stratton et al., 2009*; *Melo et al., 2022*; *Zillmann et al., 2013*; *Gómez-Molina et al., 2017*). *Noakes, Myburgh & Schall (1990)*, considering $V_{MAX}$ as a performance predictor in trained runners, also associated this role to the $V_{2VT}$. Furthermore, *Gómez-Molina et al. (2017)* and *Stratton et al. (2009)* substantiate this association by incorporating $V_{2VT}$ to $V_{MAX}$ in the multiple linear regression equation, resulting in a significant enhancement in the prediction of the performance output in trained runners. The association between $V_{2VT}$

and $V_{MAX}$ can be justified as both variables represent the aerobic and anaerobic maximal capacities of the athletes, respectively (*Gómez-Molina et al., 2017*). Higher mitochondrial density and enzymatic activity lead to lower metabolic acidosis, allowing runners to cover greater distances or increase running speed without compromising the oxidative metabolism (*Bassett & Howley, 2000*; *Gómez-Molina et al., 2017*).

In addition, increases in running speed during long-distance events are common, mainly at the beginning and in the final sprint of the race (*Maroński, 1996*). However, a higher fat percentage in recreational runners is not favorable to increase the $V_{MAX}$ values. Notably, very low-fat mass has been observed in African elite runners, who exhibit both low muscle volume and a low-fat percentage, contributing to a lower total body mass. This characteristic allows them to reach higher speeds throughout long-distance running events (*Zillmann et al., 2013*). This perspective aligns with previous research by *Hoogkamer et al. (2016)*, where they demonstrated that a 100 g increase in body mass results in approximately a 1% decrease in the 3,000 m running performance. Concerning long-distance events, *Zillmann et al. (2013)* supported the findings of the present study, demonstrating a negative association between fat percentage and $V_{MAX}$. Therefore, the fat percentage seems to be determining factor for optimal running performance in this population, as $V_{MAX}$ is directly affected.

The multiple linear regression test did not show an association between variables related to the elastic mechanism and metabolic cost in the present study. Initial variations in stride length appear to be fundamental for achieving higher speeds, while stride frequency becomes crucial at high speeds (*Novacheck, 1998*; *Peyré-Tartaruga et al., 2021*). However, the negative association between stride length and contact time may be confounded by variations in speed (*Gómez-Molina et al., 2017*), as changes in these variables are known to occur to increase running speed. Therefore, our results may not have shown associations due to the relatively slow speed used in the present study (10 km.h$^{-1}$).

The spring-mass model during running can be characterized by two major variables associated with the leg and the vertical stiffness of the center of mass ($K_{LEG}$ and $K_{VERT}$, respectively) (*Morin et al., 2005*), and these variables appear to be influenced by speed and performance level (*Carrard, Fontana & Malatesta, 2018*). *Rogers et al. (2017)* evaluated $V_{MAX}$ and $K_{LEG}$ during the 50 m sprint, along with tendon stiffness, $K_{VERT}$ (during unilateral and bilateral drop jump), running economy on a treadmill (12, 14, 16 and 18 km.h$^{-1}$), $K_{LEG}$ at 14 km.h$^{-1}$ and velocity associated with the maximal oxygen consumption in 11 highly trained male runners. They found large and moderate correlations between the values of $K_{LEG}$ (during the 50 m sprint) and $K_{VERT}$ (in unilateral jump) with the running economy at 14 km.h$^{-1}$ (*Rogers et al., 2017*), supporting the concept that tendon stiffness in the lower limb could influence the global stiffness models (running and jump), thereby running performance.

Regarding performance level, *Burns et al. (2021)* found that speed influences the biomechanical behavior and spring-mass characteristics in middle-distance elite (international/Olympic) versus trained (regional) with an average speed of 6.90 and 6.06 m.s$^{-1}$ for 1,500 m, respectively. The authors noted spatiotemporal and spring-mass parameter differences across running speeds between 10 to 18 km.h$^{-1}$. Elite runners

demonstrated greater stability in contact time, longer aerial time, and a lower duty factor than well-trained runners (*Burns et al., 2021*). The spring-mass characteristics in both groups showed that $F_{MAX}$ and $K_{VERT}$ increased with speed, while $K_{LEG}$ showed minimal changes, with elite runners exhibiting high values (*Burns et al., 2021*).

Contrary to these findings, our study did not reveal associations between variables related to the elastic mechanism and metabolic cost in recreationally active runners. It suggests that individuals engaging in leisurely exercise may adapt their step frequency and length to minimize vertical oscillation of the center of mass (*Tartaruga et al., 2012*) and leg deformation (*Morin et al., 2007*), optimizing the task. Therefore, it could potentially counterbalance any impairment in $F_{MAX}$ during each step due to a lower level of training (*da Rosa et al., 2019*), without significant effects on $K_{VERT}$ and $K_{LEG}$ at 10 km.h$^{-1}$.

On the other hand, a critical association was found between $HR_{2VT}$ and $V_{MAX}$ in the present study. Indeed, this finding is in line with previous results where *Abe et al. (1999)* observed a strong correlation between heart rate during the onset of blood lactate accumulation and $V_{MAX}$ in elite long-distance runners with similar performance levels. The heart rate found by *Abe et al. (1999)* may correspond to $HR_{2VT}$ in the present study, given that the exercise intensity was close to 4 mM, corresponding to the blood lactate values at the second ventilatory threshold approximately. Additionally, the values for $HR_{2VT}$ and $HR_{MAX}$ are consistent with previous findings (*Bunc et al., 1995*; *Hofmann et al., 1994*, *1997*; *Vucetić et al., 2014*; *Zacharogiannis & Farrally, 1993*). Moreover, our study found no association between $HR_{MAX}$ and $V_{MAX}$, also in line with the results reported by *Gómez-Molina et al. (2017)* and *Noakes, Myburgh & Schall (1990)*. A similar previous study has sought to relate the determinants of running velocity (*Wiecha et al., 2022*). However, the runners used in Weicha's study were at a higher level than ours, and using the approach recommended by *McKay et al. (2022)*, their athletes are considered trained athletes (tier 2) and ours recreationally active (tier 1).

The primary limitation of this study was the analysis of the biomechanical variables at a fixed speed of 10 km.h$^{-1}$. Conceptual models suggest variations in biomechanical responses with increasing running speed, including step length, step frequency, contact time, and aerial time (*Novacheck, 1998*; *Peyré-Tartaruga et al., 2021*). Another limitation was the absence of blood lactate concentration evaluation during the maximal incremental test. This evaluation would have brought $HR_{2VT}$ and $V_{2VT}$ closer to the actual second ventilatory threshold, offering better associations with results in the literature.

Conversely, our study provides a practical and valuable approach for analyzing the determinants of running performance and evaluating training development in recreationally active runners. The results can offer insights for coaches working with amateur and professional running groups, particularly those guiding inexperienced runners aiming to improve their health indicators or seeking information through science to enhance their running practice. Coaches should prioritize attention to the fat percentage, especially when using speed to control intensity (*Daniels, 2013*), preventing injuries from weight overload. Additionally, coaches can focus on improving $V_{2VT}$ to enhance the aerobic capacity of recreationally active runners, often through high-intensity

models (*Silva et al., 2017*). This procedure, in turn, can contribute to improved running performance and the development of new biomechanical features (*Daniels, 2013*). Therefore, this study provides valuable information for achieving optimal running performance ($V_{MAX}$).

Further, refining the heart rate ratio model (*Castagna, Krustrup & Póvoas, 2022*) gives a simple equation for estimating the metabolic cost during running. The only input variables were the maximal, rest and exercise heart rate. Using combustion enthalpy parameters (*Peyré-Tartaruga & Coertjens, 2018*), we are able to obtain a single equation for determining the metabolic cost of running. While these findings are encouraging, a couple of assumptions of the model have to be kept into account. Conditions where the cardiac output is not a key constraint of performance, as at supramaximal intensities (higher than $V_{MAX}$), the model presented is limited (*Lepretre, Koralsztein & Billat, 2004*). Indeed, at supramaximal intensities, glycolytic metabolism precludes the estimation of metabolic cost using only aerobic pathways.

Considering the established interactions between physiological, anthropometrical, and biomechanical variables, we suggest exploring non-linear approaches in future studies. Although not the most suitable for the present research questions, future studies could employ these methods to address the complexity of running performance. Network models, for instance, could offer valuable insights, even taking into account critical aspects as sex effects on the relationships studied here (*e.g.*, *Manchado-Gobatto et al., 2022*; *Pereira et al., 2018*). Furthermore, future studies may include additional variables contributing to determining $V_{MAX}$. Specifically, investigating biomechanical variables during the maximal incremental test, such as external and internal mechanical work, mechanical efficiency (*Peyré-Tartaruga et al., 2021*; *Peyré-Tartaruga & Coertjens, 2018*), and muscle and tendon mechanical properties, could offer further insights into the complex interplay affecting $V_{MAX}$.

## CONCLUSIONS

In conclusion, our study demonstrates that V2VT, metabolic cost, and fat-mass percentage can explain 62% of the variability in VMAX. Also, 90% of the variability in metabolic cost is accounted for by simple spatiotemporal variables (contact and aerial time, stride frequency and length, and vertical oscillation). Therefore, we concluded that recreationally active runners with higher $V_{2VT}$, lower metabolic cost, and a lower fat percentage will likely present better $V_{MAX}$ performance.

## ACKNOWLEDGEMENTS

We acknowledged all students that participated as coaches in the Project LOCOMOTION-Mechanics and Energetics of Terrestrial Locomotion/UFRGS: Jorge L. Storniolo, Jonas S. Hubner, Patricia D. Pantoja, Onécimo U.M. Melo, Tamiris S.S. Castro, Alex O. Fagundes, Jeam M. Geremia, Paulo R. da Silva, Vivian T. Müller, Marcos P.B. Masiero, Bruno Zanchi, Juliana J. Dias, Carlos R. de Castilhos, Leonardo S. Bloedow, Danielle Keller, Miguel A.C. Backes, Pedro Schons, Daniela V. Sacramento, Alberito R.

Carvalho, Araton C. Costa, Henrique B. Oliveira, Rodrigo G. da Rosa, Francisco B. Queiroz, Renan Coimbra, Eric M. Thomas and Rafael Dedavid.

### Funding

Leonardo A Peyré-Tartaruga is a recipient of productivity fellowship from CNPq-Brazil (302822/2019-4). This work was supported by the Fapergs (Call Fapergs/MS/CNPq/SESRS 03/2017-PPSUS, 17/2551001464-2; Call Decit/SCTIE/MS-CNPq-FAPERGS 08/2020–PPSUS, 21/2551-0000094-5; and Call Fapergs 02/2017 Programa Pesquisador Gaúcho–Pqg, 172551-0001) and Conselho Nacional de Pesquisa (CNPq) (Call Universal CNPq 2016, 422193/2016-0). The funders had no role in study design, data collection and analysis, decision to publish, or preparation of the manuscript.

### Grant Disclosures

The following grant information was disclosed by the authors:
CNPq-Brazil: 302822/2019-4.
Fapergs: Call Fapergs/MS/CNPq/SESRS 03/2017-PPSUS, 17/2551001464-2; Call Decit/SCTIE/MS-CNPq-FAPERGS 08/2020–PPSUS, 21/2551-0000094-5; and Call Fapergs 02/2017 Programa Pesquisador Gaúcho – Pqg, 172551-0001.
Conselho Nacional de Pesquisa (CNPq): Call Universal CNPq 2016, 422193/2016-0.

### Competing Interests

Leonardo A. Peyré-Tartaruga and Cosme F. Buzzachera are Academic Editors for PeerJ. The remaining authors declare that they have no competing interests.

### Author Contributions

- Leonardo A. Peyré-Tartaruga conceived and designed the experiments, analyzed the data, prepared figures and/or tables, authored or reviewed drafts of the article, and approved the final draft.
- Esthevan Machado performed the experiments, analyzed the data, prepared figures and/or tables, authored or reviewed drafts of the article, and approved the final draft.
- Patrick Guimarães performed the experiments, analyzed the data, prepared figures and/or tables, authored or reviewed drafts of the article, and approved the final draft.
- Edilson Borba performed the experiments, analyzed the data, prepared figures and/or tables, authored or reviewed drafts of the article, and approved the final draft.
- Marcus P. Tartaruga conceived and designed the experiments, analyzed the data, prepared figures and/or tables, authored or reviewed drafts of the article, and approved the final draft.
- Cosme F. Buzzachera performed the experiments, analyzed the data, prepared figures and/or tables, authored or reviewed drafts of the article, and approved the final draft.
- Luca Correale analyzed the data, prepared figures and/or tables, authored or reviewed drafts of the article, and approved the final draft.

- Fábio Juner Lanferdini conceived and designed the experiments, analyzed the data, prepared figures and/or tables, authored or reviewed drafts of the article, and approved the final draft.
- Edson Soares da Silva conceived and designed the experiments, performed the experiments, analyzed the data, prepared figures and/or tables, authored or reviewed drafts of the article, and approved the final draft.

## Human Ethics

The following information was supplied relating to ethical approvals (*i.e.*, approving body and any reference numbers):

The study was approved by the Research Ethics Committee of Universidade Federal do Rio Grande do Sul.

## Data Availability

The data is available at figshare: Peyré-Tartaruga, Leonardo (2023). paper dataset of Biomechanical, physiological and anthropometrical predictors of performance in recreational runners. figshare. Dataset. https://doi.org/10.6084/m9.figshare.23912724.v1.

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
