# Peer review of "Biomechanical, physiological and anthropometrical predictors of performance in recreational runners"

_PeerJ, doi:10.7717/peerj.16940_

## Round 0.1 · original submission · Major Revisions

Dear Authors,

The reviewers and I have completed our evaluation of your manuscript and recommend a major revision before re-submission.

Please review the comments and resubmit your revised manuscript.

**Language Note:** The review process has identified that the English language must be improved. PeerJ can provide language editing services - please contact us at copyediting@peerj.com for pricing (be sure to provide your manuscript number and title). Alternatively, you should make your own arrangements to improve the language quality and provide details in your response letter. – PeerJ Staff

Reviewer 1 ·

Basic reporting

No comment

Experimental design

No comment

Validity of the findings

No comment

Additional comments

I would like to express my gratitude for the opportunity to review the manuscript titled 'Biomechanical, Physiological, and Anthropometrical Predictors of Performance in Recreational Runners.' The primary aim of the manuscript is to examine the biomechanical, physiological, and anthropometrical factors influencing VMAX in recreational runners. The authors utilized an experimental design involving a sample of 15 recreational runners of both sexes to address this important question, given its relevance for endurance runners.

Before a more detailed analysis of the manuscript, I would like to discuss some general concerns. Firstly, I have reservations about the title and its alignment with the content. While VMAX is related to performance, it does not equate to performance. Therefore, the term 'performance in recreational runners' might not be the most appropriate.

Secondly, I urge the authors to enhance the theoretical foundation for the variables included in the regression model. Providing information on the variables used in the regression, as well as considering training variables, could improve the introduction. Additionally, explaining why variables like body mass and body fat were included as independent variables in the model and discussing potential reverse causality can strengthen the manuscript.

Another critical aspect pertains to the concept of a 'recreational runner.' Clarity on this term is essential throughout the manuscript, as it significantly impacts the generalization of the findings. Despite the ambiguity surrounding the concept in the scientific literature, the authors should furnish additional details about the sample characteristics, encompassing competitive level, training experience, and training specifics. This will enhance the transparency and applicability of the results (see https://doi.org/10.1123/ijspp.2021-0451; DOI: 10.1177/1941738113479763).

In terms of statistical analysis, I recommend incorporating tests for the assumptions underlying the use of regression models and outlining the actions taken to address any assumption violations. Specifically, providing more detail on the collinearity tests, including variables removed during analysis, would be beneficial. Did the authors perform adjustments by sex?

The results section is well written, but the quality of the figure is poor (especially fig 2 and fig 3). If possible, please, improve the quality.

Lastly, considering the established interactions between physiological, anthropometrical, and biomechanical variables, I propose that the authors explore non-linear approaches. Although not the most suitable for the present research questions, future studies could employ these methods to address the complexity of running performance. Network models, for instance, could offer valuable insights (see for example: https://www.frontiersin.org/articles/10.3389/fphys.2018.00843/full; https://www.ncbi.nlm.nih.gov/pmc/articles/PMC9311794/).

Reviewer 2 ·

Basic reporting

Thank you for the manuscript submitted for review. The manuscript titled "Biomechanical, physiological and anthropometrical predictors of performance in recreational runners" authored by Leonardo A Peyré-Tartaruga et all., delve into the biomechanical, physiological, and anthropometrical factors that predict performance in recreational runners. The study was approved by the Research Ethics Committee of Universidade Federal do Rio Grande do Sul. Fifteen recreational runners participated in the study, and the research design is described as a cross-sectional study following the recommendations of the STROBE checklist. The structure of the paper is typical for this type of study and includes the required sections.

Experimental design

The main hypotheses or objectives of the study are to verify the biomechanical, physiological, and anthropometrical determinants of VMAX in recreational runners.
Based on the introduction of the paper, I do not find scientific justification for some of the variables that were verified, e.g., the threshold heart rate level, where it is well known that both maximum and threshold HR levels are not determinants of sports performance or functional capacity.
Under the term recreational runners not much is known, the description of the study population should be detailed in the context of their training status, sports performance, other activities, etc. Table 1 is an aggregate for both sexes and should include separate information for women and men.
STROBE guidelines have been selectively addressed.
Eligibility criteria and the sources and methods of selection of participants should be described in detail.
The main objection is about the way in which variables are selected, there is not much information about potential confounders. Also, the method of variable selection itself is not sufficiently motivated in the introductory section of the paper.
The main limitation of the study that has a direct impact on the error rate is the size of the group. A group of 15 people, of two sexes, does not meet the basic criteria for achieving adequate statistical power. In view of the observational study, calculations should be made a priori. What methods were used to select the sample size of the study? The study does not include any validation of the models, either internally or externally, which, given the size of the group, would not have been possible.
Several issues in the organisation of the study need to be clarified, especially in view of the small population and the possibility of confounding factors. the study took place on a mechanical treadmill, were people somehow adapted to running on a treadmill? What kind of footwear did they have, was it standardised?
The methodology for biomechanical measurements is not described in sufficient detail, and apparatus was used about which accuracy may be questionable in laboratory use (Shishov N, PLoS One. 2021). How was this taken into account in the analysis, and is it known what the internal measurement error was during video recording?
The theoretical metabolic cost methodology also needs to be supplemented with information on the time periods from which calculations were made during exercise and resting/exercise measurements and their environmental conditions.
The methodology completely omits the section on CPET methodology and data analysis/explanation.
The statistical part of the study is based on regression models. Why was gender not analysed as a factor in the regression models? The effect on the economy of movement is influenced by the ratio of muscle mass and body fat (Maciejczyk et al., 2014), which can be taken into account as a variable. Furthermore, factors such as age would need to be verified. There is also a lack of allometric verification which is particularly important in the case of body mass.

Validity of the findings

Due to numerous shortcomings in the methodology of the work, assessing its correctness and the quality of the results is not possible at this moment. The work also lacks a section dedicated to limitations, and I notice many of these at this stage of the review

Additional comments

Introduction
Lines 62-64: I fail to see the link with the cited studies to this claim, this is not what the study by Boullosa et al, 2020 is about.
Line 68, the indicated percentage of body fat is one of the anthropometric indicators of body composition.
Line 80-90, Consider current literature such as Wiecha et al. 2022 where a prediction of Vmax was determined in a population of 4,000 runners, based on body composition indices and physiological responses
Lines 91-93, what is the scientific basis for these statements?
Lines 93-95, The cited paper (Cottin et al., 2007) does not address this research problem and is not suitable to support this thesis.
Lines 95-97, the same as above, the citation is not in the subject of the paper's research.
Lines 98-99, controversial statement, there are dozens of experimental studies in this field and many of systematic reviews like Philo U Saunders et al, 2004; Mooses M et al, 2017; Alvero-Cruz JR et al, 2020; Davis S et al.,2020.
Lines 253-255, this statement cannot be agreed with in accordance with the study by Wiech et al, 2022
Lines 281-283, please justify scientifically.

Reviewer 3 ·

Basic reporting

In this study, the researchers examined the factors influencing maximal running speed (VMAX) in recreational runners. They discovered that VMAX is affected by factors like heart rate, fat percentage, and theoretical metabolic cost. Additionally, running mechanics, including parameters such as contact time, flight time, stride length, stride frequency, and vertical oscillation, were found to impact theoretical metabolic costs. Nonetheless, the study should address language errors and provide justification for its aims and statements.

Experimental design

P.11, L.98, " The study's objective appears to lack appropriate alignment. While it initially states that the aim is to investigate the determinants of metabolic cost, it subsequently delves into the determinants of VMAX. It is recommended to thoroughly examine all the papers on the mechanical determinants of running" Mechanical Determinants of the U-Shaped Speed-Energy Cost of Running Relationship - PMC (nih.gov)

The method section is deficient in detailed information, particularly in the section addressing metabolic cost.

Validity of the findings

It is advisable to promote meaningful replication when there is a well-justified rationale.

Additional comments

Kindly review the grammar in the manuscript, as there are several typographical errors.
There are concerns regarding the clarity, correctness, or appropriateness of the language used in the study.

Annotated reviews are not available for download in order to protect the identity of reviewers who chose to remain anonymous.

---

## Round 0.2 · accepted · Accept

I have taken over handling this submission as the previous Academic Editor is not available.

Thank you for addressing all of the issues/concerns raised by the Reviewers. I am pleased to recommend your amended manuscript for publication.

Reviewer 1 ·

Basic reporting

No comment

Experimental design

No comment

Validity of the findings

No comment

Additional comments

No comment

Reviewer 3 ·

Basic reporting

No comment.

Experimental design

No comment.

Validity of the findings

No comment.

Additional comments

No comment.